# Are physiological and ecosystem -level tipping points caused by ocean acidification? A critical evaluation.

Christopher E. Cornwall[1,*], Steeve Comeau[2,*], Ben P. Harvey[3,*]

[1] School of Biological Sciences and Coastal People Southern Skies Centre of Research Excellence, Victoria University of Wellington, Wellington, 6012, New Zealand
[2] Sorbonne Université, CNRS-INSU, Laboratoire d'Océanographie de Villefranche, F–06230 Villefranche-sur-Mer, France
[3] Shimoda Marine Research Center, University of Tsukuba, Shimoda, Shizuoka, Japan
[*]These authors contributed equally.
*Correspondence to*: Christopher E. Cornwall (Christopher.cornwall@vuw.ac.nz)

**Abstract.**

Ocean acidification is predicted to cause profound shifts in many marine ecosystems by impairing the ability of calcareous taxa to calcify and grow, and by influencing the physiology of many others. In both calcifying and non-calcifying taxa, ocean acidification could further impair the ability of marine life to regulate internal pH, and thus metabolic function and/or behaviour. Identifying tipping points at which these effects will occur for different taxa due to the direct impacts of ocean acidification on organism physiology is difficult because they have not adequately been determined for most taxa, nor for ecosystems at higher levels. This is due to the presence of both resistant and sensitive species within most taxa. However, calcifying taxa such as coralline algae, corals, molluscs, and sea urchins appear to be most sensitive to ocean acidification. Conversely, non-calcareous seaweeds, seagrasses, diatoms, cephalopods, and fish tend to be more resistant, or even benefit from the direct effects of ocean acidification, though the effects of ocean acidification are more subtle for these taxa. While physiological tipping points of the effects of ocean acidification either do not exist or are not well defined, their direct effects on organism physiology will have flow on indirect effects. These indirect effects will cause ecologically tipping points in the future through changes in competition, herbivory and predation. Evidence for indirect effects and ecological change is mostly taken from benthic ecosystems in warm temperate–tropical locations *in situ* that have elevated $CO_2$. Species abundances at these locations indicate a shift away from calcifying taxa and towards non-calcareous at high $CO_2$ concentrations. For example, lower abundance of corals and coralline algae, and higher covers of non-calcareous macroalgae, often turfing species, are often found at elevated $CO_2$. However, there are some locations where only minor changes, or no detectable change occurs. Where ecological tipping points do occur, it is usually at locations with naturally elevated mean $pCO_2$ concentrations of 500 μatm or more, which also corresponds to just under that concentrations where the direct physiological impacts of ocean acidification are detectable on the most sensitive taxa in laboratory research (coralline algae and corals). Collectively, the available data support the concern that ocean acidification will most likely cause ecological change in the near future in most benthic marine ecosystems, with tipping points in some ecosystems at as low as 500 μatm $pCO_2$. However, further research is required to more adequately quantify and model the extent of these impacts in order to accurately project future marine ecosystem tipping points under ocean acidification.

# 1 Introduction

Ocean acidification is the process of increasing absorption of atmospheric $CO_2$ by the surface seawaters, leading to a decrease in pH and shift in the speciation of dissolved inorganic carbon (DIC). As a result, future seawater concentrations of $HCO_3^-$, $H^+$, and $CO_2$ will be higher, while $CO_3^{2-}$ will be lower. These changes in seawater carbonate chemistry will have complex biological consequences, as all four parameters mentioned previously hold physiological significance for various marine taxa (Hurd et al., 2019). The direct effects of ocean acidification manifest mostly through three mechanisms: 1) negatively impacting calcification (and recruitment of calcareous organisms); 2) altering photo-physiology; and 3) affecting acid-base regulation (i.e., internal pH regulation) or the energy expended in this process. These effects are usually subtle and have relatively minor direct negative effects when compared to stressors like marine heatwaves or others that can cause rapid change from one alternative stable state into another after a tipping point is reached. A tipping point is a threshold beyond which non-linear decline in ecosystem state (Van Nes et al., 2016; Carrier-Belleau et al., 2022) or physiological rates occurs (Scheffer, 2010). Specific definitions of tipping points define them as "causing a qualitative change [significantly larger than the standard deviation of natural variability" (Armstrong Mckay et al., 2022; Lenton et al., 2008). Figure 1 highlights that for some taxa this can being to occur at as little as 538 µatm $pCO_2$ (e.g., coralline algae), but for the majority of taxa we cannot be certain that such tipping points may be reached until ~ 930 µatm, though possible could occur as early as 538 as well (see Table 2 that details where these data were obtained).

While most responses to ocean acidification are linear, the cumulative impacts of these physiological changes on individual species may result in more significant ecological effects. When these effects manifest measurable physiological impacts, ecological tipping points *in situ* will likely occur. Comprehensive meta-analyses on the impacts of ocean acidification on biological processes and ecological outcomes support this concept (Leung et al., 2022; Kroeker et al., 2013b). However, these analyses rarely assess changes in processes at specific time points or at $pCO_2$ values corresponding to specific time points in the future. Instead, they usually determine whether standardised responses are different from zero. Thus, in this study, we evaluate when key responses in different taxa would differ from zero and provide $pCO_2$ values of when this typically occurs, if it is possible to make such statements given the variability in responses observed and the effects of other environmental drivers, differential effects over life history stages, and local adaptation. We also provide background information describing the key physiological and ecological impacts of ocean acidification and how linear physiological effects of ocean acidification at the organism scale can cause non-linear ecological tipping points to manifest. It must be noted here with the caveat that ocean acidification will not occur in isolation, but rather its impacts will be overlayed on top of the effects of other stressors, such as marine heatwaves, which are presently changing the abundance and distribution of many species, including creating

phase shifts between one ecosystem type and another (Wernberg et al., 2016). Those effects are beyond the prevue of this review.

## 2 Direct impacts on key physiological processes

### 2. 1 Calcification

Calcium carbonate minerals thermodynamically precipitate or dissolve relative to their saturation state ($\Omega$), which is determined by [$Ca^{2+}$] multiplied by [$CO_3^{2-}$] and divided by $K$sp. $K$sp, being dependent on seawater temperature, salinity, and pressure (Morse and Mackenzie, 1990). Consequently, under ocean acidification, declines in [$CO_3^{2-}$] and $\Omega$ lead to slower mineral precipitation in seawater. Marine organisms generally precipitate calcium carbonate under the form of calcite, aragonite, or high Mg Calcite. The choice of the mineral form is taxon and/or species specific. For a given set of conditions, calcite has higher $\Omega$ values than aragonite, while aragonite has a higher $\Omega$ than high Mg calcite (Andersson et al., 2008; Bathurst, 1972; Feely et al., 2004). Initial experimental work demonstrated that simulated ocean acidification reduces calcification in most taxa (Ries et al., 2009). However, there are also instances of resistant species/genera (e.g., *Porites* spp. corals) (Comeau et al., 2019; Fabricius et al., 2011) or entire taxa (e.g., Crustacea)(Kroeker et al., 2013b; Leung et al., 2022; Figure 1). Thus, the initial assumption was largely that ocean acidification reduces calcification by decreasing $\Omega$. Indeed, taxa that precipitated lower $\Omega$ minerals tended to be more strongly impacted by ocean acidification (Kroeker et al., 2013b). [$CO_3^{2-}$] was more highly correlated than other components of seawater carbonate chemistry in experiments where the independent role of pH, [$HCO_3^-$] and/or [$CO_3^{2-}$] were tested, and teased apart independently (Comeau et al., 2013a; Schneider and Erez, 2006). However, calcification generally occurs internally (Ries, 2011). In organisms such as corals, it takes place several tissue layers deep (Allemand et al., 2004), while in coralline algae, it occurs within the cell wall, one to several cells deep (Cornwall et al., 2017a; Mccoy et al., 2023) and in mollusc it occurs in the extrapallial space, isolated by the calcifying epithelium and the periostracum (Gazeau et al., 2013). This internal location is referred to as the calcifying fluid here, as there are a multitude of names used even within specific taxon. Notably, there are no known biological transporters of $CO_3^{2-}$, and it remains membrane impermeable in all organisms. Therefore, it is more likely that increases in external [$H^+$] drive the observed declines in calcification rates observed in corals, a process that can be offset in some taxa by the provision of greater DIC under ocean acidification (Jokiel, 2013; Comeau et al., 2018).

To calcify, marine organisms must create conditions that favour the precipitation of calcium carbonate at the site of calcification. Most calcifying organisms actively adjust their internal chemistry to initiate the inorganic precipitation of calcium carbonate by increasing the pH and ensuring a consistent supply of essential ions required for calcification ($CO_3^{2-}$, $HCO_3^-$, $Ca^{2+}$, $Mg^{2+}$) (Mcculloch et al., 2017; Decarlo et al., 2018). This is done by actively removing protons from the site of

calcification via specific cellular transporter. The inorganic carbon utilized for calcification originates from a combination of metabolic $CO_2$, $HCO_3^-$ transported to the site of calcification via cellular transporters, and ions transported via paracellular pathways (e.g., Venn et al., 2020). Additionally, marine organisms have developed specific proteins that aid in maintaining elevated calcium carbonate precipitation (Marin et al., 2007; Drake et al., 2013). While the general principles of calcification are similar across many taxa, the response to ocean acidification is highly taxa and species-specific. Here we will focus on three of the most studied calcifying taxa: corals, molluscs, and calcifying macroalgae. These three taxa are also identified as the most at risk from the effects of ocean acidification (Figure 1).

Corals have received significant attention in studies related to ocean acidification. However, results regarding a potential tipping point in their response to ocean acidification are inconclusive, and the shape of the relationship (i.e., 'reaction norms', in some fields) between calcification and pH (or $p$CO$_2$) is still a subject of debate. Early research suggested a linear response of corals growth to pH (Doney et al., 2009; Anthony et al., 2008), but some studies challenged this finding, showing non-linear relationships between calcification and pH (or $\Omega$) (e.g., Ries et al., 2009). It is however important to note that most of these studies had limited treatment levels and numbers of species, making it difficult to assess the shape of the ocean acidification to calcification relationship and detect tipping points across entire taxa. More recent studies, using a wider range of treatments (Comeau et al., 2013b, 2014) and species (Okazaki et al., 2017), experimentally demonstrated that the response of coral calcification to $p$CO$_2$ is generally linear. Linear relationships have been considered the best representation of the coral response of corals to ocean acidification in recent meta-analyses (Cornwall et al., 2021; Kornder et al., 2018). Therefore, there is no clear non-linear tipping point in the response of corals to ocean acidification, as their response is generally linear, highly species-specific, and influenced by other factors such as light, temperature, and feeding. At the reef level, linear declines in calcification occur with declines in saturation state, but the magnitude of this decrease (i.e., the slope) varies among reefs and likely depends on community composition and other drivers such as temperature. Some studies project a shift to net dissolution when $\Omega_A$ decreases below 2, a level that could be reached by the end of the century (Andersson and Gledhill, 2013; Albright et al., 2018).

Molluscs, especially commercial ones, have also been extensively studied in relation to their response to ocean acidification due to their economic importance. Because calcification and internal pH are being impacted by ocean acidification, these effects are slightly less complex than say photo-physiological effects described below. This is because it is likely two different parameters of carbonate chemistry are driving responses of molluscs to ocean acidification ($[H^+]$ and $[CO_3^{2-}]$), whereas in photosynthetic species $CO_2$ and $HCO_3^-$ additionally contribute. Thus, previous work on molluscs has often described 'reaction norms', response curves that are well defined and repeatable responses to drivers. In the context of ocean acidification, when non-linear reaction norms occur, and the threshold is within the range of carbonate chemistry predicted to occur under ocean acidification, these would also be classified as tipping points (e.g., abnormal larval % in Ventura et al., 2016 but not mortality).

However, there is s till considerable debate as to whether these reaction norms exist, even in molluscs. Previous reviews on the effects of ocean acidification on molluscs, though conducted several years ago, revealed that the response of adults and juveniles varies significantly among species and even within species (Gazeau et al., 2013; Parker et al., 2013). Similar to corals, the response of molluscs to ocean acidification is strongly influence by their environmental history and interactions with other environmental factors (Falkenberg et al., 2019; Thomsen and Melzner, 2010). Different mollusc species precipitate various forms of calcium carbonate, such as calcite, aragonite, or high Mg-Calcite, which are known to respond differently to ocean acidification. Moreover, there is evidence suggesting that molluscs could undergo transgenerational acclimation to ocean acidification, potentially mitigating the negative effects of decreasing pH (Parker et al., 2021). As a result, determining a tipping point at which molluscs calcification and growth might collapse proves to be extremely challenging. Instead, it is more likely that some species will experience a steady decrease in calcification and shell strength, while others may thrive in lower pH conditions. For example, where tipping points have been assessed, they can be far beyond the range of seawater pH values expected under ocean acidification in the coming centuries (e.g., < pH 7; Lutier et al., 2022; pH 6.62 in Caillon et al., 2023). The complexity of these responses makes it difficult to predict the overall impact of OA on molluscs with certainty.

Coralline algal are extremely sensitive to the effects of ocean acidification, with their calcification rates, cover in the field at natural $CO_2$ sites, recruitment in laboratory and in situ work, and internal pH at the site of calcification all negatively impacted by ocean acidification (Cornwall et al., 2022). Calcification rates of coralline algae are negatively affected 75% of the time, with in most instance a linear decrease of calcification with decreasing pH. However, like in molluscs, greater resistance to ocean acidification has been demonstrated in coralline algae grown for multiple generations (6 generations) under conditions simulating the effects of ocean acidification, versus those from controls (Cornwall et al., 2020; Moore et al., 2021). Additionally, some species display natural tolerance to ocean acidification (Cornwall et al., 2018; Cornwall et al., 2017a). Although the physiological mechanisms via which this tolerance is achieved is relatively unexplored, there is possible links between greater upregulation of pH within the site of calcification and maintenance of calcification under ocean acidification. Irrespective of these tolerant species, it is likely coralline algae will be among the first taxa to clearly show negative impacts of ocean acidification, as on average they are universally sensitive. This will cause large flow on effects at ecological levels (see section below on this), as they play important roles in maintaining coral reef growth and cementation (Cornwall et al., 2023), and act in many ecosystems as crucial settlement substrates for invertebrate larvae such as corals, abalone and sea urchins (Roberts, 2001; Fabricius et al., 2017).

Declines in calcification within most taxa due to ocean acidification is expected, but determining a specific tipping point for when this decline will occur is a complex task. The complexity arises from the diverse approaches used in past research, the different atmospheric $CO_2$ concentrations projected under various climate change scenarios, and the lack of alignment between meta-analyses and these atmospheric $CO_2$ concentration scenarios. We summarise the past attempts at meta-analyses, or pseudo-meta-analyses in Figure 1. Notably, there is a significant overlap between the majority of past meta-analyses (Hendriks

et al., 2010; Kroeker et al., 2013b; Kroeker et al., 2010; Harvey et al., 2013; Kornder et al., 2018) and more recent meta-analyses (Leung et al., 2022; Cornwall et al., 2021; Cornwall et al., 2022). Leung et al. offer a more detailed description of the responses of various calcifying taxa to ocean acidification compared to most prior meta-analyses. However, they group all relevant studies necessary to identify tipping points into a single category (~pH 7.9 to pH 7.6). Additionally, they group physiologically different calcifying algae that are not grouped in some other recent analyses on coralline algae [and corals], which attempt to determine temporal differences in effect sizes (Cornwall et al., 2021; Cornwall et al., 2022). Furthermore, Wittmann and Pörtner (2013), although not a true meta-analysis, provide detailed and interesting breakdowns of the impacts of different pH range values on calcification rates of various animal taxa. To estimate calcification and growth rates of most taxa, we consider these two parameters largely interchangeable. This is because the ability to create new skeletal material is essential for inorganic growth and size increases. Therefore, for our estimates, we draw on all recent relevant research, including the work of Wittmann and Pörtner.

Evidence strongly suggests that the calcification rates of coccolithophores, pteropods, coralline algae, *Halimeda* spp., corals, bivalves, and gastropods are likely to experience significant decreases due to ocean acidification by the end of the century under RCP8.5 emissions scenarios or similar conditions (e.g., ~pH 7.63 and 936 µatm $CO_2$) (Leung et al., 2022; Cornwall et al., 2022). However, the evidence remains unclear for foraminifera, calcifying sponges, bryozoans, crustaceans, polychaetes, and echinoderms, as there is mixed evidence presently available (Leung et al., 2022). Specifically, coralline algal calcification rates significantly decline from present-day rates within ocean acidification scenarios that simulate RCP4.5 in 2050 and above (e.g., 538 µatm), whereas corals experience similar departures around RCP8.5 projections for the year 2050 (e.g., 572 µatm) (Cornwall et al., 2021). Though not equivalent, the point at which more than 50% of studies find negative impacts on all measures of organism fitness is around 780 to 840 µatm for molluscs and echinoderms respectively, and above 2000 µatm for crustaceans (Wittmann and Pörtner, 2013). Although not all of these measurements directly involve calcification, this highlights the difficulty in determining physiological tipping points for changes in calcification rates for specific taxa, given the available and disparate research and the range of responses observed within a taxon.

## 2.2. Photo-physiology

Most macroalgae and all phytoplankton species have the ability to directly take up $HCO_3^-$ from seawater during photosynthesis, converting it into complex sugars (Raven et al., 2011; Raven et al., 2002a). Similarly, all marine phototrophs can passively take up $CO_2$ through diffusion for the same process (Raven et al., 2012), with some of them relying solely on this diffusive uptake. In corals, symbiotic dinoflagellates use DIC within the coral tissues through the same process (Raven et al., 2020). The active uptake of $HCO_3^-$ using pumps, symports, antiports is referred to as a $CO_2$ concentrating mechanism (CCM) (Raven et al., 2011). The evolution of CCM was believed to be a response to the low concentrations and slow diffusivity of $CO_2$ in

seawater compared to air (Raven et al., 2002b; Raven et al., 2008), where ~90% of seawater DIC is in the form of $HCO_3^-$ and only about 1% is in the form of $CO_2$. Therefore, using $HCO_3^-$ was considered to theoretically overcome DIC limitation. However, this reasoning is simplistic. This is noted in the various works by Raven and colleagues (e.g., Raven et al., 2005; Raven and Beardall, 2014, 2016), and we summarise here. There are various types of CCMs that differ in their efficiency in natural seawater. Although seawater pH typically is around ~8.05, it can be higher in regions with high photosynthetic uptake of DIC and high water retention (Rivest et al., 2017). In these habitats, $CO_2$ concentrations would be extremely small and species utilising diffusive $CO_2$ uptake would be at a competitive disadvantage over those with CCMs (Hepburn et al., 2011). In additions to true CCMs, there are various associated external and internal carbonic anhydrase enzymes that collectively enhance the diffusive $CO_2$ uptake externally, or convert internal $CO_2$ into $HCO_3^-$, thereby regulating internal $CO_2$ levels and maintaining internal pH. The efficiency and production of these enzymes can vary between species. Moreover, both the creation of CCMs and carbonic anhydrase enzymes require energy and nutrients. As a result, possessing a CCM does not guarantee that a species is not DIC limited in the habitats it currently occupies. Increasing $CO_2$ (and possibly $HCO_3^-$) could potentially alleviate DIC limitation in marine phototrophs that are constrained by DIC availability. However, quantifying the exact benefits that these organisms would receive is challenging because there has been no comprehensive effort to study this by subjecting organisms with well-known CCMs and carbonic anhydrases to seawater conditions resembling ocean acidification. Nevertheless, it has been observed that several DIC-limited species do show an increase in their photosynthetic rates and/or growth rates in response to elevated DIC or ocean acidification (Kroeker et al., 2010; Harvey et al., 2013).

Increasing $CO_2$ concentrations could also lead to species with a CCM using additional $CO_2$ when the external $CO_2$ concentration is higher, thus downregulating their CCM and reducing the energetic or nutrient cost of active $HCO_3^-$ uptake (Hepburn et al., 2011). The potential benefits of this downregulation might be minimal, but it does appear as though both marine and freshwater macroalgae at elevated $CO_2$ concentrations either downregulate their CCM or switch the ratio of $HCO_3^-$ to $CO_2$ uptake (Cornwall et al., 2017b; Maberly et al., 2014). Another possible outcome is that higher external [$CO_2$] levels could reduce the leakage of internal $CO_2$ back into seawater, saving energy and possibly nutrients. However, the specific impacts of these processes under ocean acidification are challenging to predict and likely depend on local irradiance availability and nutrient concentrations relative to the specific species' requirements. We identified two positive tipping points (coccolithophores and seagrasses) and one negative (corals) in photosynthetic responses to elevated $CO_2$. The surprising lack of responses of non-calcareous seaweeds is likely due to the complexity of their CCM presence and what suite of CCMs are present. Because some species could respond positively and others not at all, then the physiology of the species would need to be better understood in order to group them. However, there are positive tipping points in the growth rates of non-calcareous seaweeds. The meta-analyses that make up the majority of these findings are more than a decade old (Harvey et al., 2013;

Kroeker et al., 2013b), and further analyses are required that separate out seaweeds based on their CCM physiology rather than grouping them all together.

Photosynthetic efficiency (Fv/Fm) is observed to decline in many taxa (e.g., corals, seaweeds) under ocean acidification (Leung et al., 2022; Cornwall et al., 2022), but the significance of these decreases as stress indicators is debatable. For instance, declines in Fv/Fm of approximately 0.4 or more are typically associated with acute heat stress (e.g., Schoepf et al., 2019), whereas declines caused by simulated ocean acidification are usually on the order of 0.02. Therefore, it is possible that these declines are in response to non-stress related increases in cellular $CO_2$ or decreased pH, affecting the functioning of the photosystem II. Whereas acute heat stress in corals represents a well-established tipping point (mass coral bleaching), the smaller declines in Fv/Fm associated with ocean acidification may not represent an abrupt physiological tipping point, instead suggesting a gradual decline in performance, or represent a consequence of other physiological responses to the altered carbonate chemistry. Subsequently, further research is needed to understand the physiological and molecular mechanisms that underpin these responses in various phototrophic organisms and to determine if they indeed represent significant changes that would impact future photosynthetic species in a future high $CO_2$ ocean.

Meta-analyses highlight that ocean acidification will impact the photo-physiology of marine species. However, due to the complexity of the underlying processes (as highlighted above), there is considerable variability within and between taxa, and limited data on photosynthetic responses. Consequently, it becomes even more challenging to pinpoint a tipping point at which changes in photosynthetic rates would occur. On average, ocean acidification seems to have a positive impact on coccolithophore photosynthetic rates, but there is no average detectable effect in foraminifera, symbiont bearing sponges, corals, and calcifying algae (Leung et al., 2022). However, both corals and coralline algae show declines in photosynthetic efficiency, and there is evidence of both strong positive and negative impacts on their photosynthetic rates (Leung et al., 2022; Cornwall et al., 2022). The underlying physiological mechanisms for these negative observations require further investigation. Older meta-analyses also indicate positive effects of ocean acidification on seagrass, diatom, and non-calcareous macroalgal growth and photosynthesis, though with varied responses (see Figure 1) (Harvey et al., 2013; Kroeker et al., 2013b). Tipping points at which physiological changes occur in physiologically distinct groups of macroalgae and corals cannot be presently determined, and more work is required to better define the physiological function of organisms used within ocean acidification research.

## 2.3 Internal pH regulation

Ocean acidification also presents significant changes for the regulation of internal pH in different tissues or fluids of many calcifying taxa. This is evident in corals (Mcculloch et al., 2012; Venn et al., 2013), coralline algae (Cornwall et al., 2017a),
*Halimeda* spp. (Comeau et al., 2019), echinoderms (Stumpp et al., 2012), and crustaceans (Carter et al., 2013) for example. However, some of these species show an increased ability to upregulate internal pH at sites of calcification, allowing them to maintain calcification under simulated ocean acidification conditions. As a result, relying solely on meta-analyses to predict when specific species will be impacted by ocean acidification may have limitations. That being said, meta-analyses do indicate significant impacts of OA on pH regulation in certain echinoderms and corals (Leung et al., 2022), as well as in coralline algae
(Leung et al., 2022; Cornwall et al., 2022). However, the extent to which OA affects pH within different components of various taxa remains unexplored, and this could potentially explain some of the variability in responses even among non-calcifying taxa, such as many seaweeds. This is a vital controller of organism metabolism and fitness, but further research on this topic is required before tipping points could be identified. Additionally, data on internal pH within non-calcifying tissues and within non-calcifying organisms is presently limited, but is also likely strongly impacted by reduced seawater pH, and could be
driving some reduced growth rates observed in non-calcifying taxa (e.g., Taise et al. in press). Bednaršek et al. (2021) identify ~pH 7.6 as a tipping point whereby internal pH of echinoderms is impacted, but indicate that this was a subjective point determined by a panel of experts. In corals, linear effects are usually observed (Mcculloch et al., 2012; Comeau et al., 2019), and detectable differences are likely driven by a combination of the level of biological variability and the magnitude of difference between control and lower seawater pH treatments, rather than true tipping points per se. This is similar to most
responses to ocean acidification, whereby physiological tipping point identified here do not function as true ecological tipping points, but rather linear decreases. Thus, it could be expected that most physiological responses to ocean acidification would occur slowly over time in the near future.

**2.4 Whole organisms tipping points** These declines in organism physiology detailed above do not necessarily indicate that ecological tipping points would occur, and that ecosystem change will result from physiological tipping points being reached.
Rather, they indicate when change is likely to be detectable. True tipping points due to ocean acidification will likely manifest at ecological levels far before they manifest at physiological levels. For example, multiple small effects, difficult to detect using presently existing methodologies, will likely manifest across a suite of individuals, populations and species that inhabit any particular ecosystem. When these effects manifest at physiological levels, they will add up at ecological levels to much greater impacts that are much more easily detectable (Figure 3). This is concerning, as the vast majority of data we have on
some taxa is from laboratory experimentation on individuals. Equally concerning, is the issue that changes in ecological processes has likely already occurred in marine ecosystems due to ongoing ocean acidification. However, these effects are almost impossible to attribute to, due to the overall paucity of carbonate chemistry time series in most locations compared to time series in other parameters, such as seawater temperature, light, etc., and due to the subtle effects of ocean acidification in some taxa. Similarly, changes in temperature and light can be observed or felt by observers, whereas changes seawater

carbonate chemistry cannot. In the following section we discuss tipping points that have manifested in ecosystems with naturally elevated $pCO_2$ compared to nearby control sites. However, we add the caveat here that many ecosystem types remain underexplored in terms of how tipping points caused by ocean acidification could manifest.

## 3. Tipping points at naturally high CO₂ locations

Research on the potential effects of ocean acidification on marine organisms highlight the connection between increasing
atmospheric and oceanic $CO_2$ levels, changes in carbonate chemistry, and their impacts on marine organisms. As highlighted previously, numerous experimental studies have documented significant impacts of future ocean acidification on diverse aspects of individual species' physiology, life history, and ecology, as well as on populations. However, when attempting to assess the impact of ocean acidification at higher levels of biological organisation, such as community structure, food web dynamics, and ecosystem function, uncertainty arises. Particular difficulties arise when attempting to predict the outcomes of
interactions among numerous species. Nevertheless, all these individual responses contribute to ecosystem change, and so ocean acidification is anticipated to lead towards changes in the structure, composition, and functioning of marine ecosystems.

Ecosystem changes, although usually gradual, can reach tipping points where they undergo sudden and significant shifts, leading to alterations in the structure and function of biological communities. These regime shifts are of particular concern due
to the fact that the newly established habitats are primarily composed of species that have lower ecological, functional, and human value when compared to the habitats they have replaced. Examples of this include the extensive displacement of key foundation species (e.g., coral reefs and kelp forests) by simplified degraded ones like turf algae-dominated systems (e.g., Knowlton, 1992; Moy and Christie, 2012; Wernberg et al., 2016). Ecosystem changes associated with climate change are evident today in various ecosystems and biogeographic regions, often associated with temperature. However, these changes
are often complex, involving concurrent shifts in multiple environmental factors and processes. On the other hand, the direct attribution of ongoing ecosystem changes, or specific elements, to anthropogenic ocean acidification is far more difficult to establish.

Seasonal upwelling regions, such as the California coast, offer valuable insights into the contemporary impacts of altered
carbonate chemistry. In these areas, nutrient-rich waters from deeper layers of the ocean, which are also characterised by high $CO_2$ and low pH, are transported to the surface. This process provides examples of how low pH waters can bring about ecosystem changes by impacting early life stages of economically and ecologically important species (e.g., Pacific Shellfish species) (Feely et al., 2008). Another contemporary example includes coral reefs in the Equatorial Upwelling System, such as those found in the Galápagos and Cocos Islands, which exhibit low species diversity and limited development of carbonate
reef frameworks attributed to the low pH/aragonite saturation levels in the upwelling waters of the region (Glynn, 2001; Manzello, 2010; Manzello et al., 2008). One of the issues in the attribution of ecosystem change to ocean acidification (beyond

the confounding effects of other environmental factors) is associated with the difficulty in the large-scale monitoring of carbonate chemistry. However, a study examining artificial ocean alkalinisation on a localised scale yielded important findings (Albright et al., 2016). It showed that increasing alkalinity to pre-industrial levels resulted in an increase in net community calcification at a coral reef on One Tree Island, Great Barrier Reef. This study therefore suggests that contemporary communities are experiencing a decline in net community calcification compared to pre-industrial conditions, indicating that coral reef growth may potentially already be impaired by ocean acidification. Conversely, the addition of $CO_2$ to a coral reef at the same location caused declines in net community calcification rates beyond those expected to occur due to the direct impacts on coral (Albright et al., 2018). This could be due to resident coralline algae or carbonate sediments, that are high in abundance in some areas on this reef and more prone to dissolution at rates much higher than their calcification rates. While contemporary examples, such as regions of seasonal upwelling, promote our understanding of the current impacts of altered carbonate chemistry, understanding the potential future impacts of ocean acidification remains crucial.

Research efforts are also currently underway into the long-term effects of ocean acidification on ecosystem-level processes using natural analogues (Figure 2). Natural analogues for ocean acidification predominantly involve a natural, localised change in carbonate chemistry conditions that mimic future ocean acidification conditions which can be compared with an adjacent region under contemporary conditions. These include $CO_2$ seeps, associated with volcanic $CO_2$ emissions, and semi-enclosed bays (e.g., semi-enclosed lagoons, and mangrove estuaries), where certain ecologically significant parameters for marine life (carbonate chemistry, temperature, oxygen) closely resemble or even surpass the projected conditions anticipated for the end-of-the-century. Since marine communities and ecosystems are home to a wide range of species, that each interact with each other, these research approaches importantly enable researchers to simultaneously examine the response of a naturally-assembled community as a whole; regardless of whether they are investigating a specific population, functional group or the community as a whole. While not exact replicas of future conditions, these natural laboratories present unique opportunities to investigate the mechanisms by which ecosystems may adapt and respond to the challenges of climate change.

In early studies utilising $CO_2$ seeps (Hall-Spencer et al., 2008), the loss of vulnerable calcifying organisms and the potential benefit observed in certain non-calcified primary producers (such as seagrass and algae) were noted. However, it was also observed that some calcifying organisms managed to survive while certain non-calcifying primary producers were being lost. This significant finding emphasised the limitations of solely considering the direct physiological impacts of ocean acidification when predicting community-level responses, highlighting the need for a comprehensive understanding of the ecological interactions within a community (Fabricius et al., 2011; Fabricius et al., 2017). Ongoing research at a numerous $CO_2$ seeps has consistently documented shifts in habitat structure and community composition along the natural $pCO_2$ gradients. These shifts often favored simplified systems with reduced biodiversity and less ecological complexity over larger habitat-forming species (Sunday et al., 2016; Teixidó et al., 2018). Examples include transitions from canopy-forming brown algae to turf (Ischia, Italy), hard corals to soft corals (Iwotorishima, Japan), corals and macroalgae to turf (Shikine Island, Japan) (Inoue et al., 2013;

Hall-Spencer et al., 2008; Agostini et al., 2018). Many of these shifts occurred at $p$CO$_2$ conditions projected by the mid-century and suggest that thresholds of tipping points may be reached in the near future. We project that in the future, many ecosystems (but not all) will shift from states dominated by the cover of marine forests or corals to more depauperate states.

It is clearly important to distinguish whether these documented shifts in habitat structure and community composition along natural $p$CO$_2$ gradients are occurring through a tipping-point threshold being exceeded, or representing a more gradual shift along the $p$CO$_2$ gradient. The general ranges of $p$CO$_2$ at which these community shifts are observed is shown in Figure 2. For some locations, such as in the CO$_2$ seeps of Shikine Island (Japan) and Maug Islands (Commonwealth of the Northern Mariana Islands), the abrupt change in the community is first observed at sites with only slightly elevated $p$CO$_2$ conditions (450–500

µatm) (Agostini et al., 2018; Enochs et al., 2015), suggesting an ecological tipping point. Moreover, positive feedback loops, hysteresis, and altered recovery mechanisms are associated with the shift in community states at Shikine Island, further demonstrating CO$_2$-driven alternative stable states (Hudson et al., 2023; Harvey et al., 2021). However, the ecosystem shifts are first observed at higher $p$CO$_2$ conditions in other naturally-high CO$_2$ analogues (Figure 2). This does not preclude the possibility of ecological tipping points in these locations, but highlights the need for additional sites to be studied that are

closer to contemporary $p$CO$_2$ conditions in order to establish whether they represent an abrupt or gradual change in community composition.

The changes in the extent and structural complexity of biogenic habitat observed within CO$_2$ seeps can additionally mediate further biodiversity shifts, with the impact of ocean acidification on habitat-forming species projected to lead to lower

associated species diversity (Sunday et al., 2016). Field surveys in a temperate Pacific CO$_2$ seep found that the ocean acidification-driven transition in habitat resulted in a reduction in the diversity of associated fish species, this loss is largely attributed to the disappearance of species that are closely associated with habitats that have undergone significant loss themselves. Moreover, it led to the selection of fish species that are more adapted to simplified ecosystems dominated by algae (Cattano et al., 2016). Similar declines in reef-associated macroinvertebrate communities, as well as small cryptic

invertebrates, have also been observed in tropical CO$_2$ seeps (Fabricius et al., 2014; Plaisance et al., 2021).

Habitats play a crucial role in shaping the diversity of associated organisms, including macroinvertebrates and fishes; however, it is important to note that these organisms themselves also exert control over the habitats they inhabit through their grazing. In acidified conditions, it has been suggested that the top-down control exerted by some grazers may diminish, as evidenced

by reductions in the diversity, abundance and size of many marine fauna observed at CO$_2$ seeps (Garilli et al., 2015; Harvey et al., 2018; Harvey et al., 2016). For example, there is a decrease in the number of sea urchin feeding halos at a Mediterranean CO$_2$ seeps (Kroeker et al., 2013a). Top-down control in ecological systems functions as a feedback loop, where the abundance and behaviour of grazers play a crucial role in shaping the population dynamics and composition of the habitat. With any loss or reduction of grazers and their top-down control, there is a potential for significant alterations in the overall structure and

functioning of an ecosystem, and so diminished top-down control is likely contributing towards the simplification of coastal ecosystems.

The early-stage recruitment and trajectory of community development are important mechanisms responsible for altering shallow marine communities exposed to ocean acidification. Ocean acidification has notable effects on the composition of
380 prokaryotic biofilm communities on both natural and artificial substrates deployed in $CO_2$ seeps (Kerfahi et al., 2014; Taylor et al., 2014). Similarly, studies on eukaryotic biofilm communities in these seep environments have also observed changes in community composition (Allen et al., 2021). Collectively, these findings indicate that the ecological patterns driving biofilm community responses in $CO_2$ seeps are characterised by a selection for distinct cohorts of organisms, where the conditions at the reference and elevated $pCO_2$ sites represent distinct niches. When examining the longer-term development of communities,
the divergence of community composition over time is often influenced by changes in competitive interactions among habitat-forming organisms (Kroeker et al., 2013a; Kroeker et al., 2011). In general, dominant species tend to outcompete others during the early stages of succession, rapidly outgrowing or overgrowing them, which can result in the community being locked into a depauperate and low-complexity state.

When examining the impact of ocean acidification on ecosystem change, the interplay of physiological thresholds and
390 ecological interactions, such as habitat provisioning, community development, and top-down control, becomes crucial. The absence or loss of certain species indicates that either their physiology and/or their ecological dynamics are unable to cope with the adverse effects of ocean acidification. For instance, even if a species has the physiological capacity to tolerate ocean acidification conditions, it may still be competitively excluded during community development, leading to its absence in elevated $pCO_2$ conditions. For example, if crucial settlement substrates such as coralline algae are absent, ocean acidification
may then have large indirect negative impacts on corals, irrespective of direct effects (Fabricius et al., 2015; Fabricius et al., 2011; Fabricius et al., 2017). However, our understanding of the physiological mechanisms that drive these changes are poorly understood in most cases, and measurements of direct competition are often lacking in both *in situ* and laboratory research. In many natural analogues, the documented ecological shifts in the community composition often appear to be occurring at $pCO_2$ levels that are far lower than might be expected based on lethal thresholds and whole organism tipping points in isolation.
These changes occur roughly at the same $pCO_2$ where detectable physiological effects occur in laboratory experiments. We explore and demonstrate this concept in Figure 3. This mismatch is partly because the mechanisms that determine their physiological response (e.g., photophysiology, calcification) and ecological outcomes (e.g., competitive ability) are not necessarily always aligned, but are associated with how the physiological and ecological responses of a species are both important in determining their relative response within a community. For example, small changes in many species' physiology
can alter ecological interactions, with larger consequences at the ecosystem level than what would be predicted from single-

species laboratory experiments. Some of the next steps are therefore to further build upon establishing how universal it is that these ecological shifts are being caused through tipping points, rather than gradual changes.

Individual $CO_2$ seeps, while analogues for specific factors like $CO_2$, may not fully represent the broader influences of climate change. Subsequently, in recent years, research has moved towards replicating research across multiple $CO_2$ seeps (to encompass multiple environmental drivers alongside ocean acidification) (Cornwall et al., 2017b; Comeau et al., 2022), as well as making use of other natural analogues like semi-enclosed lagoons (e.g., Palau and New Caledonia), which have provided important insights into corals that have adapted to these systems (Kurihara et al., 2021; Tanvet et al., 2023). Additionally, laboratory mesocosms largely support the findings of $CO_2$ seeps, but they have both advantages and disadvantages over making inferences from them compared to $CO_2$ seeps. They can assist with determining the underlying ecological or physiological processes that will occur under ocean acidification. However, their findings are largely dictated by the number of species included, their abundances, and the study duration. Changing one of these parameters can lead to findings that are almost completely the opposite, e.g., when additional trophic levels are included or removed from an experiment, it will alter other trophic levels (Ghedini and Connell, 2016; Ghedini et al., 2015).

## 4. Conclusions

Ocean acidification has complex biological consequences, affecting calcification, photo-physiology, and pH regulation in various marine taxa. These direct impacts have been quantified in more than one thousand research articles to date. However, more research is required on internal pH regulation of various taxa (see Figure 1), and up to date meta-analyses of non-calcareous taxa (see table 2) is also required. Quantitative projections of the tipping points at which CO2 will have negative (or positive) impacts is also required for most taxa, where here we generally rely on semi-qualitative assessments for all taxa except corals and coralline algae. These negative and positive impacts will have flow on effects at ecological levels. Natural analogues, like $CO_2$ seeps, offer insights into the long-term effects of ocean acidification on marine ecosystems. These natural laboratories show shifts in habitat structure and community composition along $pCO_2$ gradients, leading to potential biodiversity loss and ecosystem simplification. However, tipping points in ecological change are difficult to determine for most taxa due to the lack of repeatable field sites with different distinct $pCO_2$ conditions. E.g., only one site presently exists for all cool temperate kelp forest ecosystems, lacking the classic gradient approach that would be useful in determining tipping points. Overall, ocean acidification has complex and far-reaching effects on marine life, and predicting specific tipping points for different taxa is challenging due to the multitude of factors involved. Continued research and monitoring efforts are essential to comprehend and address the impacts of ocean acidification on marine ecosystems in the future.

## Acknowledgements

CEC was funded by a Rutherford Discovery Fellowship from Te Apārangi The Royal Society of New Zealand (VUW 1701) and Coastal People Southern Skies Centre of Research Excellence. BPH was funded by a Japan Society for the Promotion of Science (JSPS), KAKENHI Grant-in-Aid (Grant no: 23H02231). This work contributes towards the International $CO_2$ Natural Analogues (ICONA) Network.

## Conflicts of interest

The contact author has declared that none of the authors has any competing interests.

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

| Taxon | Calcification/growth | Photosynthetic rates | Internal pH regulation |
|---|---|---|---|
| Coralline algae | ~538 | | ~538 to 930 |
| Corals | ~572 | ~538 to 930 | ~538 to 930 |
| Molluscs | ~781 | | |
| Sea urchins | ~870 | | |
| Polychaetes | ~538 to 930 | | |
| Foraminifera | ~538 to 930 | | |
| Coccolithophores | ~538 to 930 | ~538 to 930 | |
| Sponges | | | |
| Echinoderms (excl. sea urchins) | | | ~538 to 930 |
| Halimeda spp. | | | |
| Bryozoans | | | |
| Crustaceans | | | |
| Seagrasses | | ~538 to 930 | |
| Cephalopods | ~538 to 930 | | |
| Fish | ~538 to 930 | | |
| Diatoms | ~538 to 930 | | |
| Non-calcareous seaweeds | ~538 to 930 | | |

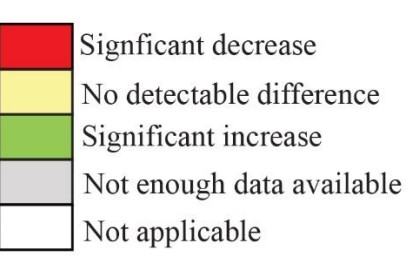

Signficant decrease
No detectable difference
Significant increase
Not enough data available
Not applicable

**Figure 1: Summary of the impacts of ocean acidification on calcification/growth, photosynthetic rates and internal pH regulation, with tipping points from literature identified where possible**. Top to bottom rows, taxa are listed from first to be negatively impacted at the top to most positively impacted at the bottom. Red = significant decrease in values from zero, yellow = no detectable difference, green = significant increase in values from zero, grey = not enough data to determine. Where possible, quantitative or qualitative tipping points given in μatm $CO_2$. Empty cells = not applicable. See Table 1 for further details on references used to generate tipping points.

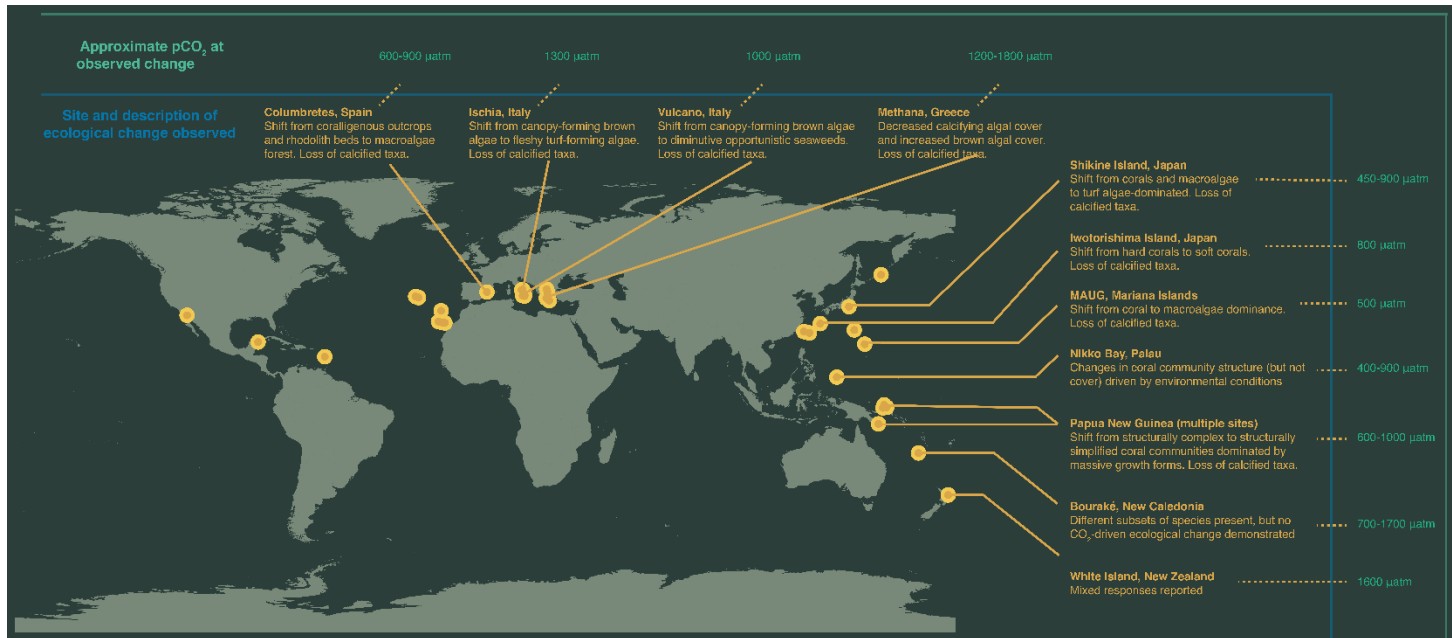

**Figure 2: Map of locations of high CO2 locations used to make inferences regarding impacts of OA on future marine ecosystems.** Labelled are sites where patterns in ecosystem change is described at Columbretes, Spain (Linares et al., 2015); Ischia, Italy (Hall-Spencer et al., 2008; Porzio et al., 2011); Vulcano, Italy (Cornwall et al., 2017b); Methana, Greece (Baggini et al., 2014); Shikine Island, Japan (Agostini et al., 2018); Iwotorishima Island, Japan (Inoue et al., 2013); Maug, Mariana Islands (Enochs et al., 2015); Papua New Guinea (Fabricius et al., 2011; Comeau et al., 2022); Bouraké, New Caledonia (Maggioni et al., 2021); White Island, New Zealand (Nagelkerken et al., 2016; Blain et al., 2021).

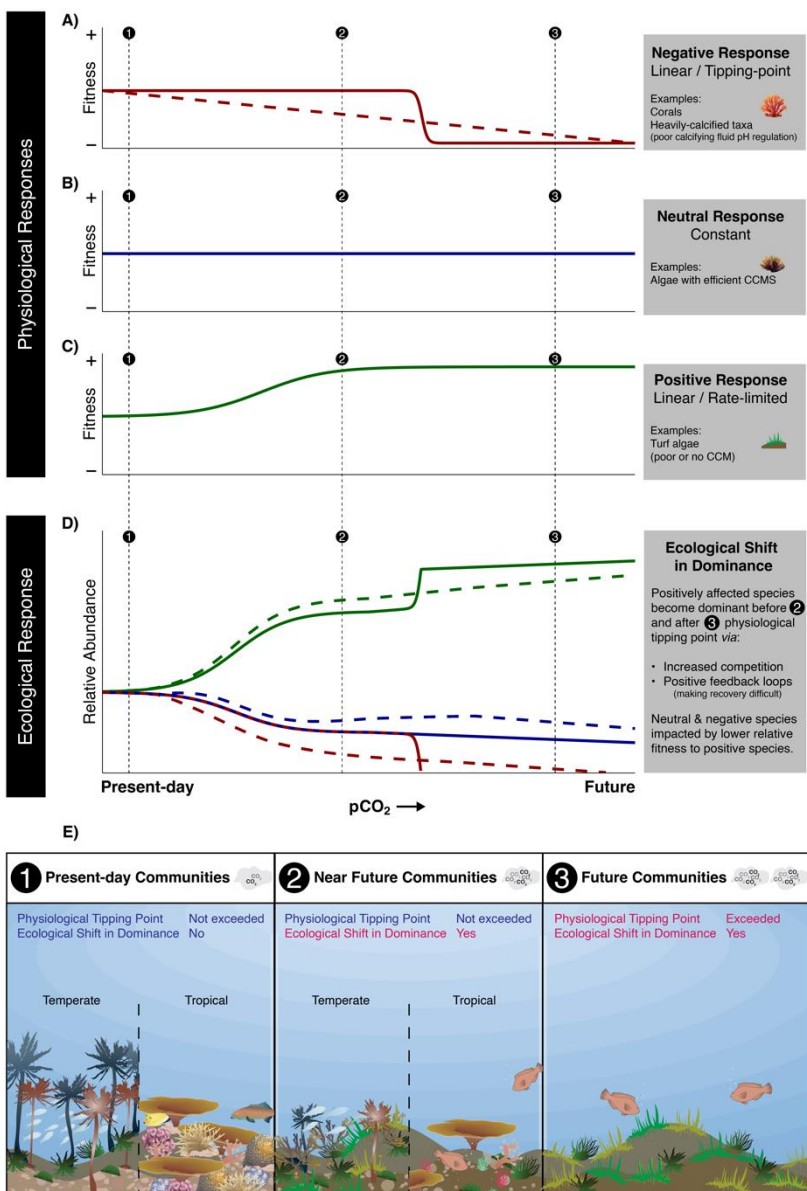

**Figure 3: Conceptual diagram demonstrating the need for understanding both physiological and ecological responses when inferring the impacts of OA on future marine ecosystems.** Species that have negative (A; solid line, tipping point; dashed line, linear), neutral (B), or positive (C) physiological responses to elevated $p$CO$_2$ conditions in isolation, may not demonstrate the same response in terms of their ecological response (D) when considered in terms of the whole community (solid line, negative response species with tipping point; dashed line, negative response species with linear response). For example, a species may physiologically have neutral responses but still show reduced relative abundance under elevated $p$CO$_2$ conditions due to alterations in their ecological interactions with other species. Subsequently, ecological shifts in dominance (ecological tipping points) may occur sooner than physiological thresholds are exceeded (E).

**Table 1: Meta-analyses or pseudo-meta-analyses that support the results of Figure 1.** Note: some studies excluded here due to merging of different taxa, inclusion of too few studies, or assessing only specific processes in combination with ocean acidification (e.g., Rivest et al., 2017; Ramajo et al., 2016). Studies listed chronologically and marked with an asterix if their results not used due to complete overlap with more recent research. Note: fish behaviour excluded from these analyses. Meta-analyses checked by searching "ocean acidification" AND "meta-analysis" on July 2023. All non-regional assessments including photosynthesis, calcification or internal pH included below.

| Study | Limited to specific taxa? | Year literature search conducted | Number of studies used | Taxa used to create Figure 1 |
|---|---|---|---|---|
| Leung (2022) | Calcifying taxa | 2020 | 985 | All calcifying taxa except *Halimeda* spp. |
| Cornwall et al. (2022) | Coralline algae | 2021 | 64 | Coralline algae |
| Schubert et al. (2023) | *Halimeda* spp. | Unknown | 31 | *Halimeda* spp. |
| Cornwall et al. (2021) | Coralline algae, corals, bioerosion, sediments | 2021 | 98 | Coralline algae and corals |
| Bednaršek et al. (2021)* | Echinoderms | 2018 | 16 | None |
| Bednaršek et al. (2019)* | Pteropods | 2017 | 15 | None. |
| Kornder et al. (2018)* | Corals | 2016 | 62 | None – though this study was used extensively in Cornwall et al. (2021) |
| Meyer and Riebesell (2015) | Coccolithophores | Unknown but 2015 or before | 33 | Coccolithophores (however, they often compared against 280 ppm $CO_2$) |
| Harvey et al. (2013) | Calcifying algae, corals, crustaceans, | 2012 | 107 | All non-calcareous taxa except diatoms |

| | echinoderms, molluscs, phytoplankton, fish, non-calcareous algae, seagrass | | | |
|---|---|---|---|---|
| Kroeker et al. (2013b) | Calcifying algae, corals, coccolithophores, molluscs, echinoderms, crustaceans, fish, fleshy algae, seagrasses, diatoms | 2012 | 155 | All non-calcareous taxa |
| Wittmann and Pörtner (2013) | Crustaceans, echinoderms, corals, fish | 2012 | 167 | Echinoderms and molluscs |
| Chan and Connolly (2013)* | Corals | 2011 | 25 | None |
| Kroeker et al. (2010)* | Calcifying algae, corals, coccolithophores, molluscs, echinoderms, crustaceans, fish, fleshy algae, seagrasses, diatoms | 2010 | 139 | None |
| Hendriks et al. (2010)* | Bivalves, corals, coccolithophores, phytoplankton, cyanobacteria | 2010 | 59 | None |

745