# Peer review of "Are physiological and ecosystem -level tipping points caused by ocean acidification? A critical evaluation."

_Earth System Dynamics, 2023_

## Referee Comment (RC1)

**Comments to the Author(s)**

This manuscript is ambitious and seeks to connect "Physiological and ecological tipping points caused by ocean acidification," as its title indicates. To my knowledge, this question has never been explored in ocean acidification research field and is of crucial importance for understanding the occurrence of tipping points and their consequences for ecosystem and global earth system functioning. This review addresses research requirements highlighted in the latest International Pannel on Climate Change's reports and is of major interest for the scientific community. By connecting tipping points in the biosphere to those occurring in the ecosphere, this study is perfectly in the scope of the journal "Earth System Dynamics".

However, despite the critical interest of its research question, this article requires major revisions before being considered for publication. First, despite its title, the article fails to discuss the link between physiological tipping points and their ecological counterpart. To address this problem, efforts must be made to connect the second part, "direct impact on key physiological processes", and the third part, "changes at naturally high $CO_2$ locations". Key concepts discussed in the article (physiological tipping point, ecological tipping point, reaction norms) are not defined in the introduction, which threatens the understanding of the entire article. There is a lack of a whole body of literature (see references below) on physiological tipping points to ocean acidification, particularly for molluscs, echinoderms, and pteropods. Discussion of tipping points is almost absent in parts 2.2 and 2.3, which contrasts with the title of the manuscript. Part 3 on "Ecological Tipping Points" lacks important concepts such as "engineer species" to effectively link with Part 2 on "Physiological Tipping Points". A figure is missing to help readers understand the ecological concepts described in Part 3. Finally, despite its crucial interest for the study, Table 1 is not really discussed even if it could help to resolve some of the problems mentioned above.

If significant revision work is done, I believe this review could be of great interest to readers of "Earth System Dynamics" and to the scientific community working on climate change in general.

**General comments**

The term "photo-physiology" has an elusive and very broad meaning. Here, the use of the term "photosynthesis" would be more relevant and would unambiguously qualify the physiological phenomenon mentioned in this article.

**Introduction**

Lines 44-46: "*However, these analyses rarely assess changes in processes at specific time points or at pCO2 values corresponding to such time points. Instead, they usually determine whether standardised responses are different from zero.*" Why is it a problem not to analyse changes in processes at specific times or pCO2 values? I think it lacks a bit of context for the reader to understand the importance of studying changes in processes over a wide range of acidified conditions (pCO2, pH, Ω), i.e. reaction norms, in order to identify tipping points.

Key concepts for the understanding of the article and its relevance should be defined in the introduction: "physiological tipping point" and "ecological tipping point". What is the definition of a tipping point for a living organism? What is the definition of a tipping point for an ecosystem? What is their relevance for the acclimation/adaptation of species to climate change, the functioning of the Earth system and conservation/mitigation policies? Why is it so important to identify them? These concepts are never clearly defined in the article threatening the global understanding. This reference can help: *Carrier-Belleau et al., 2022. "Tipping Points and Multiple Drivers in Changing Aquatic Ecosystems: A Review of Experimental Studies." Limnology and Oceanography 67, no. S1: S312–30.*

Moreover, I think the article would benefit from defining the concept of "reaction norms" in the introduction. Indeed, establishing reaction norms (modelling physiological processes over a wide acidification range) is essential to identify tipping points. Reaction norms can take many different forms and can for example be non-linear, e.g. showing tipping points, or linear. This explains the different reaction norms observed in corals mentioned by the authors in lines 79-83. Therefore, defining the concept of "reaction norms" would greatly improve clarity for the reader. This should be done in the introduction when the authors discuss the different methodologies used in ocean acidification research.

**Part 2. Direct impact on key physiological processes**

Lines 56-57. "*However, there are also instances of resistant species or entire taxa (Kroeker et al., 2013b; Leung et al., 2022).*" Please specify the names of these taxa.

Lines 77-78. "*These three taxa are also identified as the most at risk from the effects of ocean acidification.*" Please add a reference.

Line 80. "*relationship.*" Replace with "reaction norm". Same comment for the rest of the article.

The paragraph, lines 94-105, on tipping points in mollusc calcification, omits references available in the literature referring to this issue. The following references must be included:
> Bednaršek et al., 2019*. "Systematic Review and Meta-Analysis Toward Synthesis of Thresholds of Ocean Acidification Impacts on Calcifying Pteropods and Interactions With Warming." Frontiers in Marine Science 6.* This meta-analysis identify tipping points for the whole taxon of pteropods including for calcification and shell structure.
> Lutier et al., 2022. *"Revisiting Tolerance to Ocean Acidification: Insights from a New Framework Combining Physiological and Molecular Tipping Points of Pacific Oyster." Global Change Biology 28, no. 10.* This study describes tipping points in the calcification and shell parameters of the pearl oyster.

Lines 120-135. This paragraph regrets the lack of studies determining the physiological tipping point in the response of species/taxa to ocean acidification, and in particular the lack of meta-analyses. Although rare, such studies does exist. There is, for example, two meta-analyses (Bednaršek et al., 2021, 2019). It is therefore necessary to cite more references so as not to give a false impression to the reader, for example (this list is non-exhaustive):
> Bednaršek et al., 2021*. "Synthesis of Thresholds of Ocean Acidification Impacts on Echinoderms." Frontiers in Marine Science 8.*
> Dorey et al., 2013 *"Assessing Physiological Tipping Point of Sea Urchin Larvae Exposed to a Broad Range of pH." Global Change Biology 19, no. 11.*
> Lee et al., 2019. *"Tipping Points of Gastric pH Regulation and Energetics in the Sea Urchin Larva Exposed to CO2 -Induced Seawater Acidification." Comparative Biochemistry and Physiology Part A: Molecular & Integrative Physiology 234 (August 1, 2019): 87–97.*
> Lutier et al., 2023. "Pacific Oysters Do Not Compensate Growth Retardation Following Extreme Acidification Events." Biology Letters 19, no. 8
> Ventura et al., 2016. *"Maintained Larval Growth in Mussel Larvae Exposed to Acidified Under-Saturated Seawater." Scientific Reports 6, no. 1.*
> Bamber, 1990. *"The Effects of Acidic Seawater on Three Species of Lamellibranch Mollusc." Journal of Experimental Marine Biology and Ecology 143*
> Bamber, 1987. *"The Effects of Acidic Sea Water on Young Carpet-Shell Clams Venerupis Decussata (L.) (Mollusca: Veneracea)." Journal of Experimental Marine Biology and Ecology 108, no. 3*

There is almost no discussion of "tipping points" in subsections 2.2 and 2.3 on "photophysiology" and "internal pH regulation" even though this concept is emphasized in the title of the article. I understand that this is due to a lack of literature on the subject. However, I think it is possible to make assumptions about the existence of potential tipping points in these key physiological processes based on all the information presents in literature. For example, some thresholds seems to exist in the functioning of photosynthesis in organisms exposed to ocean acidification (Chen et al., 2014. *"A Red Tide Alga Grown under Ocean Acidification Upregulates Its Tolerance to Lower pH by Increasing Its Photophysiological Functions."*). For tipping points in "internal pH regulation", data are presents in Bednaršek et al. (2021) and Lee et al. (2019).

**Part 3. Changes at naturally high $CO_2$ locations**

I think the title of this part should be changed to "ecological tipping points" for clarity. Indeed, the study of changes in an ecosystem naturally rich in $CO_2$ is only one of the methodologies for determining tipping points, another being modelling. Further efforts should be made to define what ecosystem tipping points are, i.e. the transition from a stable to an unstable state, and why it is important to identify them (conservation policies, etc.). A conceptual figure would benefit the reader and greatly improve clarity.

Additional efforts should also be made to connect physiological tipping points with ecosystem tipping points. Indeed, for the moment part 3 is not really linked to part 2. The questions are: "What happens to a species when ocean acidification reaches its physiological tipping point?" "How does this translate into tipping points for the ecosystem? ». The authors are already exploring the modification of competition between species by ocean acidification in an attempt to link the physiology of species to ecology. However, this needs to be explored further, for example by discussing the key concepts of "engineer species", "trophic food webs" and "cascade effect". Some mesocosm experiments could also provide clues to the existence of ecological tipping points, for example:

> Legrand et al., 2017. *"Species Interactions Can Shift the Response of a Maerl Bed Community to Ocean Acidification and Warming." Biogeosciences 14, no. 23*
> Wright et al., 2018. *"Ocean Acidification Affects Both the Predator and Prey to Alter Interactions between the Oyster Crassostrea Gigas (Thunberg, 1793) and the Whelk Tenguella Marginalba (Blainville, 1832)." Marine Biology 165, no. 3*

This reference could also help:

> Monaco & Helmuth, 2011. *"Chapter Three - Tipping Points, Thresholds and the Keystone Role of Physiology in Marine Climate Change Research." In Advances in Marine Biology, Academic Press, 2011.*

**Conclusion**

Lines 326-328. *"Quantitative projections of the tipping points at which CO2 will have negative (or positive) impacts is also required for most taxa, where here we generally rely on semi qualitative assessments for all taxa except corals and coralline algae."* Incorrect, see reference list for echinoderms and molluscs above.

---

## Referee Comment (RC2)

Review comments:

General comments

This work presents a great depiction of the current state of understanding of the impact of OA on organisms and community structure. The text is generally direct and clearly written. The text generally suggests that identifying tipping points caused by OA is difficult and there are other factors to be considered. My major concern is that the paper reads more as a summary of the impact of OA rather than an assessment of how tipping point can be caused by OA.

The first parts (section 2) have some paragraphs that are a bit clearer about the difficulty of identifying tipping points from a physiological perspective only and translating this to any broader ecological processes.

From my point of view, the other parts however seem mainly about changes/shifts and do not touch enough on how these changes are likely to occur; gradual (linear) vs abrupt (tipping point/bi-phasic).

Another consideration is whether tipping points are likely to be caused by CO2 alone, or influenced primarily by interactions with other parameters/stressors?

Are there shifts in community structure or simplification of food web occurring through a tipping point or are they more gradual shifts along the spatial/temporal gradient of pCO2? No doubt that a shift in community structure/habitat forming species and so on are likely, and in fact shown in some areas (e.g., CO2 seep having different communities). However, can a proper sudden change in stable state be identified at a given range of pCO2 or are these occurring mostly gradually or randomly at different sites based on the site's own characteristics? I think it is important here to clearly differentiate between gradual shift vs a sudden tipping point, or maybe the lack of appropriate data/studies to properly infer tipping points.

There is also a lack of more concrete direction on how to look properly into identifying actual sudden change in stable state in natural systems, for example, or thresholds in pCO2, especially in the concluding section.

Nonetheless, in my point of view, this can be a nice addition to the literature and provide some nice insights given it revised carefully and a little restructuring. In its current state, it feels a little as a summary to the impact of OA, to which some inferences about tipping points have been made. I hope that these comments help improve the paper.

Specific questions:

The title gives the impression that OA definitely causes tipping points, which from the text seems the opposite. In addition, I believe 'ecological tipping point' is a bit vague/too broad. Perhaps the title can be modified to put forward the main message of the paper a bit more clearly.

Line 25-30: Where does the 500 come from? I mean if it is table 1 then it seems only based on some physiological processes (which is stated), not 'ecological' tipping points.

Broadly across the text: It is not clear to me what is meant by ecological tipping point, for two reasons. Firstly, it sounds very broad, and the scope of this paper is not so broad. Secondly, because it is not clearly stated, for example in the intro.

Lines 34-44: I think the introduction needs a bit more information on how OA can cause tipping points in terms of physiology, population, and community. A general introduction of the concept, so to speak. Also, there should be a clear definition of 'tipping point' against which all the studies within this paper are being judged or inferences being made. In its current state, if I were to read only the intro, I would think that this is a paper on a summary of the impact of OA on organisms. There is no information on tipping points.

Line 60-78: This section consists of detailed information on calcification. I wonder whether such detail is needed to sway the reader towards understanding how OA links with physiological tipping points? Would some of that information be better off being used within the into and links to thresholds/tipping points be made there? Or perhaps made more concise and links to tipping points made more explicit?

Lines 79-119: The studies from different taxa seem to suggest that a tipping point only caused by OA is rather unlikely and that there is a more linear decrease. Would that mean that a sudden shift in the measured responses would most likely be driven by cumulative effects of stressors rather than singular effect of pH? I understand that the context here is primarily on OA, but cumulative impacts also mean involvement of OA, albeit with other parameters/stressors (e,g., Russell, B. D., THOMPSON, J. A. I., Falkenberg, L. J., & Connell, S. D. (2009). Synergistic effects of climate change and local stressors: CO2 and nutrient-driven change in subtidal rocky habitats. Global Change Biology, 15(9), 2153-2162. *And many other references around that topic*). Therefore, would it be worth adding some information on the cumulative effects of OA and other stressors on calcification?

Lines 149-179: Same comment as the comments above about calcification (lines 60-78).

Lines 250-255: No doubt that these sites provide a great opportunity to study differences in physiology, population, and communities. One of the issues is that in many instances there is a large gap between the different pCO2 levels at which parameters are being compared (that is no proper gradient), meaning that even if a very large shift in community is observed, the way that this shift has occurred is not clear. Meaning that one cannot clearly infer whether there has been a gradual shift, a sudden tipping point, or a combination of multiple factors that have changed in a stepwise manner. Similar to what the authors pinpoint in the introduction (line 45, and also line 83, for example). Perhaps this is also a gap in many natural and lab-based studies that needs to be stressed on?

Lines 263-275: This is interesting and shows clearly that communities change in different ways, sometimes not as one would expect. Can the authors expand a bit more on the pCO2 thresholds and show the link with tipping of the communities? Or are these thresholds and links not clear enough to be called tipping points?

Line 268-275, and lines 329-331: I am wondering whether it would help to produce a diagram that shows the trends of changing community along measured pH/pCO2 values from these sites? I am thinking that this will more clearly show where the communities tip to a different/simplified state along the increasing

pCO2 (if that is the case) and whether there are clear differences between where that happens for different sites.

Line 311: remove "this" in "this ocean acidification"?

Lines 286-321: This section is a nice discussion about the changes/consequences in ecological interactions driven by OA. I wonder how this links with tipping points and thresholds in pCO2/pH? I think this is the only part missing here.

Lines 323-334: Are there a couple key gaps that can be pinpointed here regarding identifying tipping points caused by CO2 and key pathways to fill these gaps? I mean apart from better quantitative analysis/projections.

Figure 1: It is not clear whether this figure shows the actual thresholds of pCO2 that cause a tipping point or represents the general ranges of where different communities are documented in these sites. Perhaps this can be clarified in the figure and legend.

Table 2: I feel like table 2 can be supplementary. It does not seem to add too much more info to the text except as a support to table 1, which already contains references.

---

## Author Comment (AC1)

**VICTORIA UNIVERSITY OF WELLINGTON**
*Te Whare Wānanga o te Ūpoko o te Ika a Māui*

[Figure]

**Dr. Christopher E. Cornwall**
**Senior Lecturer**
**School of Biological Sciences**
**Victoria University of Wellington**

Dear Dr Steven Lade

We received the comments from two reviewers, which we address below. We thank you and the reviewers for your time in appraising our manuscript, and apologise for the misunderstanding regarding the word count limit we thought we had set on this special issue. Both reviewers provided a positive appraisal of the manuscript but had suggestions for some revisions. We now have thoroughly revised the manuscript and below we detail the revisions that we have conducted and our responses to reviewer suggestions and questions. We detail the reviewer comments in quotations and in italics, followed by our revisions and responses in plain text after each comment. Here in this attached file we address Reviewer #1's comments.

*"Reviewer #1:*

*This manuscript is ambitious and seeks to connect "Physiological and ecological tipping points caused by ocean acidification," as its title indicates. To my knowledge, this question has never been explored in ocean acidification research field and is of crucial importance for understanding the occurrence of tipping points and their consequences for ecosystem and global earth system functioning. This review addresses research requirements highlighted in the latest International Pannel on Climate Change's reports and is of major interest for the scientific community. By connecting tipping points in the biosphere to those occurring in the ecosphere, this study is perfectly in the scope of the journal "Earth System Dynamics"."*

We thank the reviewer for this positive appraisal of this manuscript.

*"However, despite the critical interest of its research question, this article requires major revisions before being considered for publication. First, despite its title, the article fails to discuss the link between physiological tipping points and their ecological counterpart. To address this problem, efforts must be made to connect the second part, "direct impact on key physiological processes", and the third part, "changes at naturally high $CO_2$ locations"."*

We now add further substantial discussion in between the second and third section of this manuscript to address the lack of linkages between section 2 on the physiology of organisms, and its third section on ecological responses (new section 2.4.).

*"Key concepts discussed in the article (physiological tipping point, ecological tipping point, reaction norms) are not defined in the introduction, which threatens the understanding of the entire article."*

We agree with the reviewer that these points should have been included. We apologise, but the reason they were not included is because we were under the mistaken impression we had a much smaller word limit, and because this submission is part of a special issue on tipping points and we assumed this would be introduced in the summary papers. We now introduce these terms for the reader.

*"There is a lack of a whole body of literature (see references in the supplement file) on physiological tipping points to ocean acidification, particularly for molluscs, echinoderms, and pteropods."*

We now add these and many others. Apologies, we had mistakenly thought we had to keep these sections very brief, so there is a body of literature on all organisms that is missing, not just those pointed out by the reviewer here.

*"Discussion of tipping points is almost absent in parts 2.2 and 2.3, which contrasts with the title of the manuscript."*

We now add linking material within these sections back to the tipping point concept.

*"Part 3 on "Ecological Tipping Points" lacks important concepts such as "engineer species" to effectively link with Part 2 on "Physiological Tipping Points". A figure is missing to help readers understand the ecological concepts described in Part 3."*

We have addressed this concern by adding further material in section 3 that introduces these concepts now, where appropriate.

*"Finally, despite its crucial interest for the study, Table 1 is not really discussed even if it could help to resolve some of the problems mentioned above."*

Table one is where we obtain the 'tipping points', where they exist, and we previously did not extensively discuss the table because of perceived word counts issues. We now discuss these extensively in each section.

*"If significant revision work is done, I believe this review could be of great interest to readers of "Earth System Dynamics" and to the scientific community working on climate change in general.*

[Figure]

**VICTORIA UNIVERSITY OF WELLINGTON**
**Te Whare Wānanga o te Ūpoko o te Ika a Māui**

*Please find suggestions for improving the manuscript in the supplement file"*

Thank you for your positive appraisal of this manuscript.

*"Attached file:*

*General comments  The term "photo-physiology" has an elusive and very broad meaning. Here, the use of the term "photosynthesis" would be more relevant and would unambiguously qualify the physiological phenomenon mentioned in this article."*

We respectfully disagree. Photosynthesis is a process that involves many processes that work in tandem (e.g., CCM operation), and most of the general readership unfortunately will mistake "photosynthesis" with "photosynthetic rates", and we consider this is dangerous to equate Fv/Fm and other related measurements to photosynthetic rates. Hence, we are actually being more specific, rather than ambiguous here in our opinion.

*"Introduction  Lines 44-46: "However, these analyses rarely assess changes in processes at specific time points or at pCO2 values corresponding to such time points. Instead, they usually determine whether standardised responses are different from zero." Why is it a problem not to analyse changes in processes at specific times or pCO2 values? I think it lacks a bit of context for the reader to understand the importance of studying changes in processes over a wide range of acidified conditions (pCO2, pH, Ω), i.e. reaction norms, in order to identify tipping points.  Key concepts for the understanding of the article and its relevance should be defined in the introduction: "physiological tipping point" and "ecological tipping point". What is the definition of a tipping point for a living organism? What is the definition of a tipping point for an ecosystem? What is their relevance for the acclimation/adaptation of species to climate change, the functioning of the Earth system and conservation/mitigation policies? Why is it so important to identify them? These concepts are never clearly defined in the article threatening the global understanding. This reference can help: Carrier-Belleau et al., 2022. "Tipping Points and Multiple Drivers in Changing Aquatic Ecosystems: A Review of Experimental Studies." Limnology and Oceanography 67, no. S1: S312–30."*

Yes, we accept that this should be introduced as part of a standalone paper on the topic. We now add this material, but leave this to the editor, as the manuscript is part of a special issue on this topic. We do not define reaction norms, as this phrase is perhaps unhelpful in understanding complex responses to ocean acidification, and we do not use it here.

*"...Moreover, I think the article would benefit from defining the concept of "reaction norms" in the introduction. Indeed, establishing reaction norms (modelling physiological processes over a wide acidification range) is essential to identify tipping points. Reaction norms can*

[Figure]

*take many different forms and can for example be non-linear, e.g. showing tipping points, or linear. This explains the different reaction norms observed in corals mentioned by the authors in lines 7983. Therefore, defining the concept of "reaction norms" would greatly improve clarity for the reader. This should be done in the introduction when the authors discuss the different methodologies used in ocean acidification research."*

We use the term reaction norm to explain the effects on molluscs, but for other taxa we respectfully disagree on including this phrase/topic. There is no reaction norm for the response of more complex organisms to ocean acidification, and we consider using this phrase to mislead junior readers. For example, concentrations of $H^+$, $CO_2$, $HCO_3^-$ and even saturation state itself can all impact the physiology of taxa that both calcify and photosynthesize. Thus, while for organisms where few physiological processes are being impacted by ocean acidification (e.g. molluscs) this concept is useful, understanding of these processes could be hampered by using such terms here. Additionally, at least in the organisms we study, environmental contexts and individual variability often modify responses beyond the usefulness of such terms.

*"Part 2. Direct impact on key physiological processes  Lines 56-57. "However, there are also instances of resistant species or entire taxa (Kroeker et al., 2013b; Leung et al., 2022)." Please specify the names of these taxa."*

Added, as per table 1 that outlines some of these (e.g., Crustaceans), a well as examples of resistant species (e.g., some *Porites* spp. corals).

*"Lines 77-78. "These three taxa are also identified as the most at risk from the effects of ocean acidification." Please add a reference."*

Added "Table 1".

*"Line 80. "relationship." Replace with "reaction norm". Same comment for the rest of the article."*

Replaced.

*"The paragraph, lines 94-105, on tipping points in mollusc calcification, omits references available in the literature referring to this issue. The following references must be included: Bednaršek et al., 2019. "Systematic Review and Meta-Analysis Toward Synthesis of Thresholds of Ocean Acidification Impacts on Calcifying Pteropods and Interactions With*

*Warming." Frontiers in Marine Science 6. This meta-analysis identify tipping points for the whole taxon of pteropods including for calcification and shell structure. Lutier et al., 2022. "Revisiting Tolerance to Ocean Acidification: Insights from a New Framework Combining Physiological and Molecular Tipping Points of Pacific Oyster. Global Change Biology 28, no. 10. This study describes tipping points in the calcification and shell parameters of the pearl oyster.""*

It was a mistake to omit Bednaršek et al. 2019 in our table, and it is now included. However, the more recent Leung et al. paper completely overlaps its findings. Bednaršek et al. 2019 also compares saturation state and not more easily comparable and physiologically relevant pH/$p$CO$_2$. Additionally, the difficult issue with that paper is that they separately analyse growth and calcification, and that the median "ocean acidification" treatments used had an aragonite saturation sate < 1 (i.e., much lower than the tipping point already identified previously for molluscs in our manuscript).

The Lutier et al. manuscript only examines one species, and they also find extremely low pH tipping points, beyond which will occur under ocean acidification (pH ~7). Thus, it is of limited utility here for our analysis. That being said, a good paper to add to our discussion, thank you for point this out. We now add a citation to this and other papers to our discussion.

*"Lines 120-135. This paragraph regrets the lack of studies determining the physiological tipping point in the response of species/taxa to ocean acidification, and in particular the lack of metaanalyses. Although rare, such studies does exist. There is, for example, two meta-analyses (Bednaršek et al., 2021, 2019). It is therefore necessary to cite more references so as not to give a false impression to the reader, for example (this list is non-exhaustive): Bednaršek et al., 2021. "Synthesis of Thresholds of Ocean Acidification Impacts on Echinoderms." Frontiers in Marine Science 8. Dorey et al., 2013 "Assessing Physiological Tipping Point of Sea Urchin Larvae Exposed to a Broad Range of pH." Global Change Biology 19, no. 11. Lee et al., 2019. "Tipping Points of Gastric pH Regulation and Energetics in the Sea Urchin Larva Exposed to CO2 -Induced Seawater Acidification." Comparative Biochemistry and Physiology Part A: Molecular & Integrative Physiology 234 (August 1, 2019): 87–97. Lutier et al., 2023. "Pacific Oysters Do Not Compensate Growth Retardation Following Extreme Acidification Events." Biology Letters 19, no. 8 Ventura et al., 2016. "Maintained Larval Growth in Mussel Larvae Exposed to Acidified Under-Saturated Seawater." Scientific Reports 6, no. 1. Bamber, 1990. "The Effects of Acidic Seawater on Three Species of Lamellibranch Mollusc." Journal of Experimental Marine Biology and Ecology 143 Bamber, 1987. "The Effects of Acidic Sea Water on Young Carpet-Shell Clams Venerupis Decussata (L.) (Mollusca: Veneracea)." Journal of Experimental Marine Biology and Ecology 108, no. 3"*

We now cite some of these papers where appropriate. We were under the false impression we had a much tighter word count than we did and we already had around 100 references. However, some of these papers discuss individual instances, and others find tipping points that are much lower than will occur due to ocean acidification (i.e., pH 7.2, pH 6.9). Thus,

there is a difference between a "pH tipping point" that will occur below pH 7, and a tipping point that will actually be caused by ocean acidification in the next 100 years.

*"There is almost no discussion of "tipping points" in subsections 2.2 and 2.3 on "photophysiology" and "internal pH regulation" even though this concept is emphasized in the title of the article. I understand that this is due to a lack of literature on the subject. However, I think it is possible to make assumptions about the existence of potential tipping points in these key physiological processes based on all the information presents in literature. For example, some thresholds seems to exist in the functioning of photosynthesis in organisms exposed to ocean acidification (Chen et al., 2014. "A Red Tide Alga Grown under Ocean Acidification Upregulates Its Tolerance to Lower pH by Increasing Its Photophysiological Functions."). For tipping points in "internal pH regulation", data are presents in Bednaršek et al. (2021) and Lee et al. (2019)."*

Physiological tipping points in internal pH cannot be found for larger groups of logical physiological groups of seaweeds and coral. Thus, we now explicitly state this for the reader.

There are hundreds of papers examining the impacts of ocean acidification on photosynthetic rates/photo-physiology (including 50+ species assessed by the author team), so unfortunately we cannot use this cited example to help the reader. We clarify that we mean there is no meta-analysis that could find tipping points. Nor could there ever with the literature the way it is. This is because photo-physiology is extremely complex, and most studies do not define the affinity of their species for DIC (nor how the CCM changes or not), so we are left with hundreds of studies with responses of organisms that physiologically would be expected not to respond to ocean acidification, and others who would be expected to respond positively. However, these responses are also strongly mediated by light, water motion and other factors often not measured.

*"Part 3. Changes at naturally high CO2 locations  I think the title of this part should be changed to "ecological tipping points" for clarity. Indeed, the study of changes in an ecosystem naturally rich in CO2 is only one of the methodologies for determining tipping points, another being modelling…"*

We thank the reviewer for their comment and change the title of this section now.

*"Further efforts should be made to define what ecosystem tipping points are, i.e. the transition from a stable to an unstable state, and why it is important to identify them (conservation policies, etc.). A conceptual figure would benefit the reader and greatly improve clarity.  Additional efforts should also be made to connect physiological tipping points with ecosystem tipping points. Indeed, for the moment part 3 is not really linked to part*

[Figure]

*2. The questions are: "What happens to a species when ocean acidification reaches its physiological tipping point?" "How does this translate into tipping points for the ecosystem? ». The authors are already exploring the modification of competition between species by ocean acidification in an attempt to link the physiology of species to ecology."*

We now add material addressing this in an entire new sub-section 2.4 as well as wording within section 3, and now add a conceptual figure. We also further clarify that the smaller, almost non-existent, physiological tipping points combine to impact all species within an ecosystem, that manifests in more easily defined and measured ecosystem shifts. Thus, while tipping points in the physiology of most species truly do not exist (outside of the effects on calcification), they do at ecological levels, where small, difficult to measure and variable effects add up to alter ecosystem functioning.

*"However, this needs to be explored further, for example by discussing the key concepts of "engineer species", "trophic food webs" and "cascade effect"…."*

We consider all readers should have learned this in their undergraduate training by now. We understand the merit of including such terms, and particularly stable state hypotheses. However, it is technically difficult to properly prove the existence of stable or unstable states in most existing ocean acidification work due to the lack of long-term time series in field studies.

*"Some mesocosm experiments could also provide clues to the existence of ecological tipping points, for example: Legrand et al., 2017. "Species Interactions Can Shift the Response of a Maerl Bed Community to Ocean Acidification and Warming." Biogeosciences 14, no. 23 Wright et al., 2018. "Ocean Acidification Affects Both the Predator and Prey to Alter Interactions between the Oyster Crassostrea Gigas (Thunberg, 1793) and the Whelk Tenguella Marginalba (Blainville, 1832)." Marine Biology 165, no. 3 This reference could also help: Monaco & Helmuth, 2011. "Chapter Three - Tipping Points, Thresholds and the Keystone Role of Physiology in Marine Climate Change Research." In Advances in Marine Biology, Academic Press, 2011""".*

A good point raised by the reviewer. We now add additional material that involves attempts to determine change in mesocosm experiments, or at least discusses information gained and the caveats.

*".  Conclusion Lines 326-328. "Quantitative projections of the tipping points at which CO2 will have negative (or positive) impacts is also required for most taxa, where here we generally rely on semi qualitative assessments for all taxa except corals and coralline algae." Incorrect, see reference list for echinoderms and molluscs above."*

**VICTORIA UNIVERSITY OF WELLINGTON**
*Te Whare Wānanga o te Ūpoko o te Ika a Māui*

[Figure]

We agree that for echinoderms and other taxa this does exist, but for "most" taxa it does not, as we state originally. Additionally, some find tipping points far below environmentally relevant seawater pH/$pCO_2$ values, or assign subjective points during linear declines. We now alter our manuscript accordingly.

---

## Author Comment (AC2)

**VICTORIA UNIVERSITY OF WELLINGTON**
*Te Whare Wānanga o te Ūpoko o te Ika a Māui*

[Figure]

Dr. Christopher E. Cornwall
Senior Lecturer
School of Biological Sciences
Victoria University of Wellington

Dear Dr Steven Lade

We received the comments from two reviewers, which we address below. We thank you and the reviewers for your time in appraising our manuscript, and apologise for the misunderstanding regarding the word count limit we thought we had set on this special issue. Both reviewers provided a positive appraisal of the manuscript but had suggestions for some revisions. We now have thoroughly revised the manuscript and below we detail the revisions that we have conducted and our responses to reviewer suggestions and questions. We detail the reviewer comments in quotations and in italics, followed by our revisions and responses in plain text after each comment. We respond to reviewer #2 in this letter.

*"Reviewer #2:*

*This work presents a great depiction of the current state of understanding of the impact of OA on organisms and community structure. The text is generally direct and clearly written. The text generally suggests that identifying tipping points caused by OA is difficult and there are other factors to be considered. My major concern is that the paper reads more as a summary of the impact of OA rather than an assessment of how tipping point can be caused by OA.*

*The first parts (section 2) have some paragraphs that are a bit clearer about the difficulty of identifying tipping points from a physiological perspective only and translating this to any broader ecological processes."*

Thank you for your time reviewing the manuscript and your positive appraisal.

*"From my point of view, the other parts however seem mainly about changes/shifts and do not touch enough on how these changes are likely to occur; gradual (linear) vs abrupt (tipping point/bi-phasic).*

*Another consideration is whether tipping points are likely to be caused by CO2 alone, or influenced primarily by interactions with other parameters/stressors?*

*Are there shifts in community structure or simplification of food web occurring through a tipping point or are they more gradual shifts along the spatial/temporal gradient of pCO2? No doubt that a shift in community structure/habitat forming species and so on are likely, and in fact shown in some areas (e.g., CO2 seep having different communities). However, can a proper sudden change in stable state be identified at a given range of pCO2 or are these occurring mostly gradually or randomly at different sites based on the site's own characteristics? I think it is important here to clearly differentiate between gradual shift vs a sudden tipping point, or maybe the lack of appropriate data/studies to properly infer tipping points."*

There are all important points that should be included in the manuscript, and we now add material in section 3 discussing these points.

*"There is also a lack of more concrete direction on how to look properly into identifying actual sudden change in stable state in natural systems, for example, or thresholds in pCO2, especially in the concluding section."*

We now add material in the discussion around this. Field manipulations of $pCO_2$ would be the only way to achieve this, though if these were fully replicated across a range of different $pCO_2$ values and also replicated in time and space, it would be logistically complex.

*"Nonetheless, in my point of view, this can be a nice addition to the literature and provide some nice insights given it revised carefully and a little restructuring. In its current state, it feels a little as a summary to the impact of OA, to which some inferences about tipping points have been made. I hope that these comments help improve the paper.*

Thank you for your positive appraisal of the manuscript.

*"Specific questions (in attached file):*

*The title gives the impression that OA definitely causes tipping points, which from the text seems the opposite. In addition, I believe 'ecological tipping point' is a bit vague/too broad. Perhaps the title can be modified to put forward the main message of the paper a bit more clearly."*

We agree, and now change the title to "Are physiological and ecosystem-level tipping points caused by ocean acidification? A critical evaluation."

*"Line 25-30: Where does the 500 come from? I mean if it is table 1 then it seems only based on some physiological processes (which is stated), not 'ecological' tipping points."*

This is from the field studies noted in Figure 1. We now detail these further in the discussion in section 3.

*"Broadly across the text: It is not clear to me what is meant by ecological tipping point, for two reasons. Firstly, it sounds very broad, and the scope of this paper is not so broad. Secondly, because it is not clearly stated, for example in the intro. Lines 34-44: I think the introduction needs a bit more information on how OA can cause tipping points in terms of physiology, population, and community. A general introduction of the concept, so to speak. Also, there should be a clear definition of 'tipping point' against which all the studies within this paper are being judged or inferences being made. In its current state, if I were to read only the intro, I would think that this is a paper on a summary of the impact of OA on organisms. There is no information on tipping points."*

We respond to the similar comment from reviewer #1, see the file in response to them for more details. Briefly, this is part of a special issue on tipping points, and thus we expect that these concepts to be properly introduced there. However, as a paper that might be read stand-alone, we now add this introduction for the reader.

*"Line 60-78: This section consists of detailed information on calcification. I wonder whether such detail is needed to sway the reader towards understanding how OA links with physiological tipping points? Would some of that information be better off being used within the into and links to thresholds/tipping points be made there? Or perhaps made more concise and links to tipping points made more explicit?"*

We now better link these processes to how physiological tipping points exist, and what responses we would expect to see due to changes in seawater carbonate chemistry.

*"Lines 79-119: The studies from different taxa seem to suggest that a tipping point only caused by OA is rather unlikely and that there is a more linear decrease. Would that mean that a sudden shift in the measured responses would most likely be driven by cumulative effects of stressors rather than singular effect of pH? I understand that the context here is primarily on OA, but cumulative impacts also mean involvement of OA, albeit with other parameters/stressors (e,g., Russell, B. D., THOMPSON, J. A. I., Falkenberg, L. J., & Connell, S. D. (2009). Synergistic effects of climate change and local stressors: CO2 and nutrient-driven change in subtidal rocky habitats. Global Change Biology, 15(9), 2153-2162. And many other references around that topic). Therefore, would it be worth adding some information on the cumulative effects of OA and other stressors on calcification?"*

We now add material that acknowledges the role OA plays in acting in tandem with other drivers. However, this is not the main aim of our manuscript, and to add this material would be complex. For example, warming can increase coral calcification rates, but only in cool seasons, whereas in warmer seasons increases in temperature will drive extensive mortality of corals.

*"Lines 149-179: Same comment as the comments above about calcification (lines 60-78)."*

Please see our reply above.

*"Lines 250-255: No doubt that these sites provide a great opportunity to study differences in physiology, population, and communities. One of the issues is that in many instances there is a large gap between the different pCO2 levels at which parameters are being compared (that is no proper gradient), meaning that even if a very large shift in community is observed, the way that this shift has occurred is not clear. Meaning that one cannot clearly infer whether there has been a gradual shift, a sudden tipping point, or a combination of multiple factors that have changed in a stepwise manner. Similar to what the authors pinpoint in the introduction (line 45, and also line 83, for example). Perhaps this is also a gap in many natural and lab-based studies that needs to be stressed on?"*

Yes, that is true for many field studies, we now add a sentence discussing this caveat. I think an issue here is also how well pCO2/pH etc is monitored at some of these locations and whether such smaller gradients in pCO2 exist consistently enough to act as natural analogues without receiving critique that they also have periods of time where pCO2 is too similar to control conditions. However, at some sites there are 100 uatm differences and also changes in coral and turf abundance already occur there (e.g., Shikine and some places in Papua New Guinea).

For laboratory work, the problems likely are in that it is difficult to control and properly manipulate seawater carbonate chemistry to precisely obtain $pCO_2$ levels that are between 450 and 500 μatm. For example, high profile papers discussing the need to conduct experiments at ~430 μatm do not realise that present day variability in most coastal zones far exceeds 430, and that experimental organisms would also have bene exposed to these levels on a regular basis. We now add this discussion in the text near the end of our conclusions.

*"Lines 263-275: This is interesting and shows clearly that communities change in different ways, sometimes not as one would expect. Can the authors expand a bit more on the pCO2 thresholds and show the link with tipping of the communities? Or are these thresholds and links not clear enough to be called tipping points?"*

We now add material full discussing this in section 3. The conceptual figure should hopefully help show that $pCO_2$ thresholds cannot necessarily explain the ecosystem level tipping points, but rather the physiology of the organisms present and their responses must also be known.

*"Line 268-275, and lines 329-331: I am wondering whether it would help to produce a diagram that shows the trends of changing community along measured pH/pCO2 values from*

*these sites? I am thinking that this will more clearly show where the communities tip to a different/simplified state along the increasingpCO2 (if that is the case) and whether there are clear differences between where that happens for different sites.*"

This would be too difficult to include here without re-analysing all of the data from each seep site. And in some instances, there are small changes in community structure (e.g., shifts between different Cystoseira spp. in Panarea) versus larger shifts in other locations (e.g., complete removal of all coralline algae at Vulcano). Thus, it is too tricky to include. But now we include a conceptual diagram that could be useful for the readers in understanding how physiological changes add up at ecological levels.

*"Line 311: remove "this" in "this ocean acidification"?"*

Amended.

*"Lines 286-321: This section is a nice discussion about the changes/consequences in ecological interactions driven by OA. I wonder how this links with tipping points and thresholds in pCO2/pH? I think this is the only part missing here."*

We now add this into the discussion here.

*"Lines 323-334: Are there a couple key gaps that can be pinpointed here regarding identifying tipping points caused by CO2 and key pathways to fill these gaps? I mean apart from better quantitative analysis/projections."*

We now add further recommendations to the manuscript. A greater physiological knowledge, how this adds to ecological outcomes, and better testing/further testing of species-specific responses across gradient type approaches in larger organisms (as has been done in phytoplankton etc for instance).

*"Figure 1: It is not clear whether this figure shows the actual thresholds of pCO2 that cause a tipping point or represents the general ranges of where different communities are documented in these sites. Perhaps this can be clarified in the figure and legend."*

This figure shows the general ranges of pCO2 where the different community shifts were observed. We now indicate this clearly in the figure legend.

**VICTORIA UNIVERSITY OF WELLINGTON**
*Te Whare Wānanga o te Ūpoko o te Ika a Māui*

[Figure]

*"Table 2: I feel like table 2 can be supplementary. It does not seem to add too much more info to the text except as a support to table 1, which already contains references."*

We are happy to move this back to the supplementary. We will leave this to the editor to decide.

---

## Author Comment (AC3)

②

③

②

③

**Futu**

pCO₂ ⟶

**ities** <small>CO₂ CO₂</small> ② **Near Future Communities** <small>CO₂ CO₂ CO₂ CO₂</small> ③ **Futur**

---

## Author Response (AR2)

**VICTORIA UNIVERSITY OF WELLINGTON**
**Te Whare Wānanga o te Ūpoko o te Ika a Māui**

[Figure]

**Dr. Christopher E. Cornwall**
**Senior Lecturer**
**School of Biological Sciences**
**Victoria University of Wellington**

Dear Dr Steven Lade                                                                                      11/03/2024

We received the editorial comments on 29/02/2024 that we address below. We thank you for your time in appraising our manuscript. We detail the comments we received in quotations and in italics below, followed by our revisions and responses in plain text after each comment.

*"I find unsatisfactory the authors' response to the comment beginning 'There is almost no discussion of "tipping points" in subsections 2.2 and 2.3 on "photophysiology" and "internal pH regulation"...'.*

*- References to tipping points in section 2.2 remain sparse, limited only to the final paragraph, despite the reviewer's repeated requests to include them. It would be worthwhile clarifying which mechanisms referenced in previous paragraph could be potential tipping points.*

*- I understand identifying the position of the tipping point is difficult given current literature, or even impossible given the extent to which the tipping point is affected by other pressures. However, I think the reviewer is saying that acknowledging there is evidence for a tipping point is useful, even if the exact position cannot be found."*

We have now modified this text further to point out where these occur.

Note that there are 11/17 identified tipping points for calcification/growth but only 3/9 for photosynthetic rates and 3/17 for internal pH. Hence our reluctance to discuss them.

*"Otherwise I find the authors' changes and responses to be appropriate. I have just a couple final minor style suggestions:*

*- To have the introduction as one long single paragraph is unusual. I suggest to break it into multiple paragraphs".*

Amended as per the editor's suggestion.

*"- Consider referring to Table 1 and especially Table 2 earlier. For example you could write "provide pCO2 values of when this typically occurs (Table 1)" in introduction"*

Amended as per the editor's suggestion. We now refer to them in the introduction.

*"- It's also unusual to have as key methodological detail as the search terms for a review buried in the caption (Table 2). I suggest you promote this to the main text."*

Amended as per the editor's suggestion.

*"- I also note the journal's comment about colours not being permitted in tables. You will need to either submit the Table as a Figure or remove colouring from the Table. Or perhaps you could design a figure that communicates the Table's information graphically and with annotations indicating the numbers and references."*

We have now turned this into a figure (Figure 1) and modified it by removing the references and adding a key.